# Adaptive Classifier-Free Guidance via Dynamic Low-Confidence Masking

**Pengxiang Li**[*1], **Shilin Yan**[*♠2], **Joey Tsai**[3], **Renrui Zhang**[4],
**Ruichuan An**[5], **Ziyu Guo**[4], **Xiaowei Gao**[†6]

[1]PolyU    [2]Alibaba    [3]THU    [4]CUHK    [5]PKU    [6]ICL

{2040gis, tattoo.ysl}@gmail.com

[*]Equal Contribution    ♠Project Leader    [†]Corresponding Author

## Abstract

Classifier-Free Guidance (CFG) significantly enhances controllability in generative models by interpolating conditional and unconditional predictions. However, standard CFG often employs a static unconditional input, which can be suboptimal for iterative generation processes where model uncertainty varies dynamically. We introduce Adaptive Classifier-Free Guidance (A-CFG), a novel method that tailors the unconditional input by leveraging the model's instantaneous predictive confidence. At each step of an iterative (masked) diffusion language model, A-CFG identifies tokens in the currently generated sequence for which the model exhibits low confidence. These tokens are temporarily re-masked to create a dynamic, localized unconditional input. This focuses CFG's corrective influence precisely on areas of ambiguity, leading to more effective guidance. We integrate A-CFG into a state-of-the-art masked diffusion language model and demonstrate its efficacy. Experiments on diverse language generation benchmarks show that A-CFG yields substantial improvements over standard CFG, achieving, for instance, a 3.9 point gain on GPQA. Our work highlights the benefit of dynamically adapting guidance mechanisms to model uncertainty in iterative generation. Code is available at https://github.com/pixeli99/A-CFG.

## 1    Introduction

Diffusion models [33, 14] have recently revolutionized generative modeling, demonstrating remarkable capabilities in synthesizing high-fidelity data in continuous domains such as image and audio [9, 29]. This success has ignited a surge of interest in extending their power to discrete data, with natural language generation standing as a particularly compelling frontier [2, 20, 11]. Among these efforts, Masked Diffusion Models (MDMs), exemplified by frameworks like LLaDA [26], have emerged as a promising direction. These models learn to reverse a gradual masking process, iteratively infilling masked tokens to construct coherent text, offering a principled and flexible alternative to traditional autoregressive language generation.

A pivotal advancement that significantly amplified the practical utility of diffusion models, especially in conditional settings, is Classifier-Free Guidance (CFG) [15]. Originally conceived for continuous models, CFG provides an elegant way to steer the generation process towards a desired conditioning signal (e.g., a textual prompt) by interpolating between conditional and unconditional model predictions during the reverse diffusion (denoising) phase. This is achieved without the need for an auxiliary classifier, making CFG a versatile and widely adopted mechanism for enhancing sample quality and

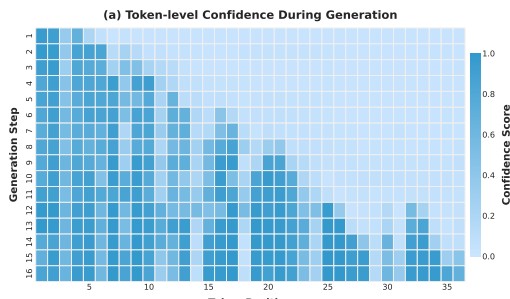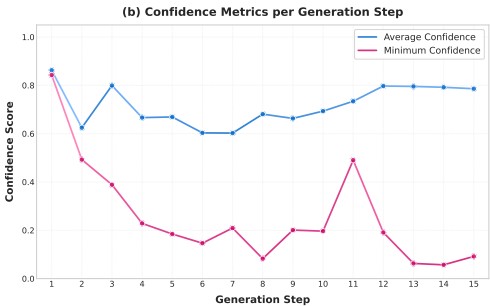

Figure 1: **Overview of model confidence dynamics during iterative generation.** (a) Token-level confidence heatmap across token positions and generation steps (darker shades indicate higher confidence). (b) Average and minimum confidence scores per generation step. This visualization highlights the dynamic and non-uniform nature of model confidence that A-CFG aims to leverage.

controllability. Naturally, the application of CFG has extended to textual diffusion models, where it plays a similar role in guiding text generation.

However, the conventional application of CFG within iterative (masked) diffusion language models often encounters a subtle yet significant limitation: the "unconditional" prediction typically relies on a *static* or *generic* construct. This often involves using a null prompt or a sequence where all target tokens are uniformly masked to simulate an unconditional state. While straightforward, such a fixed approach to unconditioning may not fully harness CFG's potential in the dynamic context of iterative text refinement. As an MLM progressively fills in a sequence, its internal state of certainty can vary considerably across different tokens and denoising steps. A static unconditional baseline fails to adapt to these nuances, potentially leading to guidance that is either too weak, too diffuse, or misaligned with the model's specific points of ambiguity at a given step.

This observation sparks a crucial question: can the "unconditional" component of CFG, when applied to iterative diffusion language models, be rendered more intelligent and responsive to the model's own evolving understanding of the sequence? We posit that the model's instantaneous predictive confidence during the iterative denoising process, **which, as visualized in Figure 1, can fluctuate significantly across tokens and generation steps,** offers a rich, yet largely untapped signal. Instead of a blanket, context-agnostic unconditioning, what if we could dynamically shape the unconditional input to reflect and address the model's current uncertainties? This would allow the guidance mechanism to concentrate its corrective influence precisely where it is most needed.

In this paper, we introduce **Adaptive Classifier-Free Guidance (A-CFG)**, a novel framework designed to realize this vision for iterative (masked) diffusion language models. A-CFG dynamically synthesizes the input for the unconditional prediction by identifying and temporarily re-masking tokens for which the conditional diffusion model exhibits low predictive confidence during a given denoising step. By doing so, A-CFG creates a localized "unconditional" state that compels the model to reconsider its predictions at these specific points of ambiguity. The standard CFG formula is then applied, leveraging this adaptively constructed unconditional state to steer the generation with greater precision and efficacy.

We integrate and evaluate A-CFG within the LLaDA [26] framework. Our extensive experiments on a range of standard language generation benchmarks demonstrate that A-CFG yields substantial improvements in **complex reasoning accuracy and adherence to conditional prompts** over both baseline LLaDA without CFG and LLaDA employing traditional CFG with static unconditional inputs. Specifically, A-CFG achieves up to a 3.9 point absolute improvement on the GPQA benchmark and enhances Sudoku task success by 8.0 points when compared to standard CFG.

Our contributions are thus threefold:

- We identify and articulate the limitations of static unconditioning in standard CFG when applied to iterative masked language models.
- We propose Adaptive Classifier-Free Guidance (A-CFG), a novel method that dynamically constructs the unconditional input based on the model's predictive confidence, enabling more targeted and effective guidance.

- We demonstrate through comprehensive experiments that A-CFG significantly enhances the performance of the LLaDA model on various generation tasks, outperforming standard CFG.

## 2 Related Work

### 2.1 Diffusion Models for Language Generation

Autoregressive (AR) models, such as large language models (LLMs) like GPT-style architectures [27, 5] and more recent powerful open-source models including LLaMA [35, 36], Qwen [6], and Mistral [18], have become the dominant paradigm in natural language generation. These models generate text token by token, conditioning each new token on the previously generated sequence, and have demonstrated remarkable capabilities across a wide array of tasks. Their success has also spurred extensions into traditional multimodal domains [17, 41, 39, 40, 37], combining language understanding with other modalities like vision [19, 16, 24, 3, 10, 1, 22, 38]. However, the sequential nature of AR generation can lead to challenges such as error propagation and limitations in bidirectional context modeling for certain tasks.

In response to these and other considerations, diffusion models [33, 14] have emerged as a powerful alternative. While initially demonstrating success in continuous domains like images [9, 29], significant effort has been dedicated to adapting them for discrete data, particularly text [2, 20, 11]. Early approaches explored discrete state-space diffusion [2] or continuous diffusion in embedding spaces (e.g., Diffusion-LM [20], DiffuSeq [11]), showcasing potential for controllability but often lagging behind AR models in likelihood or efficiency.

A particularly relevant and successful direction has been the development of **Masked Diffusion Models (MDMs)** [30, 32]. These models formulate text generation as an iterative mask-infilling process, learning to reverse a gradual masking procedure. Prominent examples like LLaDA [26] have demonstrated that MDMs can achieve competitive performance with strong AR models on various language tasks, even at scale. These models operate by iteratively refining a sequence, making them a prime candidate for fine-grained guidance techniques. Our work focuses specifically on enhancing conditional generation within such iterative, masked diffusion frameworks like LLaDA.

### 2.2 Classifier-Free Guidance in Generative Models

Classifier-Free Guidance (CFG) [15] has become a cornerstone technique for improving sample quality and conditional control in diffusion models, initially popularized in image synthesis [29]. It elegantly steers generation towards a condition by interpolating between conditional and unconditional model predictions during the reverse process, avoiding the need for separate classifier training. This is typically achieved by training the diffusion model with occasional dropout of the conditioning signal (e.g., null text prompt), enabling it to produce both conditional and unconditional outputs.

The adaptation of CFG to language diffusion models [23, 25] presents unique considerations. A common practice is to simulate the unconditional prediction by providing a static input, such as a fully masked target sequence. While effective, this static unconditioning strategy poses a limitation, particularly for iterative MDMs. As the model refines the text sequence over multiple steps, its internal state of certainty varies across different token positions and time steps. A fixed unconditional baseline fails to adapt to these dynamics, potentially leading to suboptimal or misaligned guidance.

## 3 Methodology

Our work introduces Adaptive Classifier-Free Guidance (A-CFG), a novel enhancement to the Classifier-Free Guidance (CFG) paradigm. A-CFG is specifically designed for iterative masked language models (MLMs) and aims to improve generative control by dynamically constructing the unconditional input required for CFG. This is achieved by leveraging the model's instantaneous predictive confidence regarding its current non-`[MASK]` tokens, allowing guidance to be more precisely targeted towards regions of the sequence where the model exhibits uncertainty. Figure 2 provides a high-level comparison of standard CFG with our proposed A-CFG.

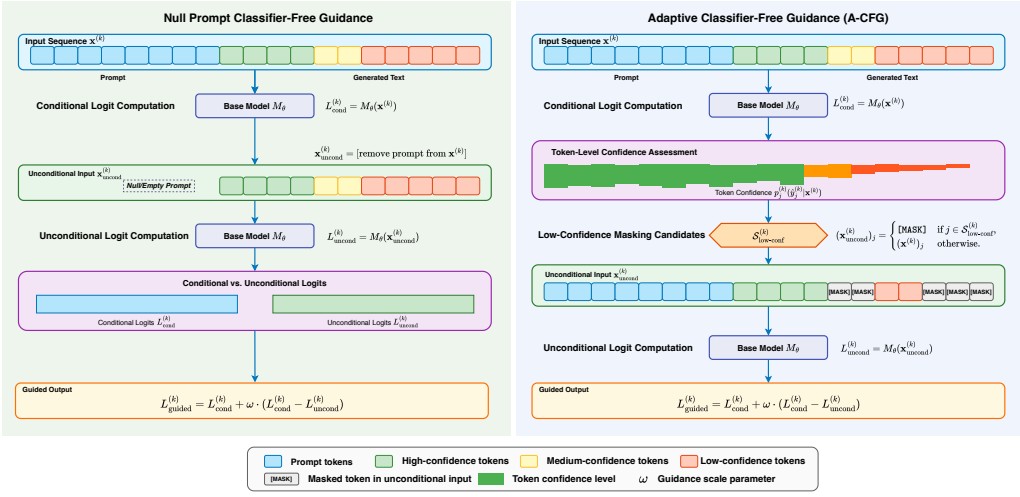

Figure 2: **Overview of (left) standard Null Prompt Classifier-Free Guidance and (right) our proposed Adaptive Classifier-Free Guidance (A-CFG) at a single generation step** $k$. **In standard CFG, the unconditional input often involves masking the entire prompt or using a null prompt.** In A-CFG, after computing conditional logits from $\mathbf{x}^{(k)}$, token-level confidences for all non-[MASK] tokens in $\mathbf{x}^{(k)}$ are assessed. Tokens with low confidence (orange/red in illustration) are temporarily re-masked to [MASK] to create the dynamic unconditional input $\mathbf{x}_{\text{uncond}}^{(k)}$. This allows the CFG mechanism to focus guidance on areas of model uncertainty within the current sequence.

## 3.1 Preliminaries

Before detailing A-CFG, we briefly review the foundational concepts: iterative masked language modeling and standard classifier-free guidance.

**Iterative Masked Language Models (MLMs).** Our A-CFG framework operates within the context of iterative generation, characteristic of many masked language models like LLaDA. Text generation commences with an input sequence $x$ that is either partially or entirely populated with special [MASK] tokens. The generation unfolds over a series of steps. At each step $k$, the model $M_\theta$ predicts replacement tokens for a subset (or all) of the extant [MASK] tokens. This iterative process progressively refines the sequence $x^{(k)}$ until a complete output $x^{(0)}$ is achieved (where $k$ typically decreases from an initial number of steps down to 0). The core predictive mechanism involves the model $M_\theta(x^{(k)})$ producing logits over the vocabulary for the positions designated for infilling.

**Classifier-Free Guidance (CFG).** Classifier-Free Guidance [15] is a widely adopted technique for enhancing sample quality and controllability in conditional generative models. CFG operates by linearly interpolating the outputs derived from a conditional model prediction, $L_{\text{cond}}(x^{(k)}, c)$, and an unconditional model prediction, $L_{\text{uncond}}(x^{(k)}, \emptyset)$. Here, $c$ represents the conditioning information (e.g., a textual prompt), and $\emptyset$ signifies a null or broadly unconditional context. The construction of this unconditional input can vary; for instance, some approaches derive an unconditional-like term from a masked version of the conditioning prompt itself [25]. The guided logits, $L_{\text{guided}}$, are computed as:

$$L_{\text{guided}}(x^{(k)}, c) = L_{\text{uncond}}(x^{(k)}, \emptyset) + (w + 1) \cdot (L_{\text{cond}}(x^{(k)}, c) - L_{\text{uncond}}(x^{(k)}, \emptyset)), \tag{1}$$

where $L$ denotes the model's output logits, and $w$ is the guidance scale. A guidance scale $w > 0$ amplifies the influence of the conditioning signal $c$. A central challenge in applying CFG, particularly in iterative MLM frameworks, is the effective definition and derivation of the unconditional logits $L_{\text{uncond}}(x^{(k)}, \emptyset)$, an issue directly addressed by our A-CFG approach.

## 3.2 Adaptive Classifier-Free Guidance (A-CFG)

Standard CFG, while effective, often relies on a static or generic definition for the unconditional prediction $L_{\text{uncond}}(x^{(k)}, \emptyset)$ when applied to iterative MLMs. Typically, this involves using a null

---

**Algorithm 1** Adaptive Classifier-Free Guidance (A-CFG) for one generation step $k$

---

1: **Input:** Current sequence $\mathbf{x}^{(k)}$, conditioning $c$, model $M_\theta$, guidance $w$, re-masking proportion $\rho$.
2: **Output:** Guided logits $L_{\text{guided}}^{(k)}$.
3: $L_{\text{cond}}^{(k)} \leftarrow M_\theta(\mathbf{x}^{(k)})$                                          $\triangleright$ Compute conditional logits
4: $\mathcal{C}_{\text{remaskable}}^{(k)} \leftarrow \{j \mid (\mathbf{x}^{(k)})_j \neq \texttt{[MASK]}\}$          $\triangleright$ Identify all non-$\texttt{[MASK]}$ token indices
5: $\mathcal{CONF}^{(k)} \leftarrow \emptyset$
6: **for** $j \in \mathcal{C}_{\text{remaskable}}^{(k)}$ **do**
7:      $c_j^{(k)} \leftarrow \max_v(\text{softmax}(L_{\text{cond}}^{(k)}))_{j,v}$           $\triangleright$ Assess confidence for remaskable tokens
8:      Add $(c_j^{(k)}, j)$ to $\mathcal{CONF}^{(k)}$
9: **end for**
10: $\mathcal{S}_{\text{low-conf}}^{(k)} \leftarrow \emptyset$
11: **if** $|\mathcal{C}_{\text{remaskable}}^{(k)}| > 0$ **then**
12:      $N_m^{\text{target}} \leftarrow \lceil \rho \cdot |\mathcal{C}_{\text{remaskable}}^{(k)}| \rceil$
13:      $N_m^{\text{actual}} \leftarrow \min(N_m^{\text{target}}, |\mathcal{C}_{\text{remaskable}}^{(k)}|)$
14:      **if** $N_m^{\text{actual}} > 0$ **then**
15:          Sort $\mathcal{CONF}^{(k)}$ by confidence values $c_j^{(k)}$ in ascending order.
16:          $\mathcal{S}_{\text{low-conf}}^{(k)} \leftarrow$ indices $j$ of the first $N_m^{\text{actual}}$ elements in sorted $\mathcal{CONF}^{(k)}$.
17:      **end if**
18: **end if**
19: $\mathbf{x}_{\text{uncond}}^{(k)} \leftarrow \mathbf{x}^{(k)}$
20: **for** $j \in \mathcal{S}_{\text{low-conf}}^{(k)}$ **do**
21:      $(\mathbf{x}_{\text{uncond}}^{(k)})_j \leftarrow \texttt{[MASK]}$                $\triangleright$ Create dynamic unconditional input
22: **end for**
23: $L_{\text{uncond}}^{(k)} \leftarrow M_\theta(\mathbf{x}_{\text{uncond}}^{(k)})$                          $\triangleright$ Compute unconditional logits
24: $L_{\text{guided}}^{(k)} \leftarrow L_{\text{uncond}}^{(k)} + (w+1) \cdot (L_{\text{cond}}^{(k)} - L_{\text{uncond}}^{(k)})$      $\triangleright$ Apply CFG formula
25: **return** $L_{\text{guided}}^{(k)}$

---

prompt or masking all prompt tokens to simulate an unconditional state. In complex generation scenarios, the model's uncertainty can fluctuate significantly. A static or predefined unconditioning strategy might therefore apply guidance indiscriminately, potentially misdirecting the generation process or failing to provide sufficient correction where it is most needed. This observation motivates A-CFG. Our core intuition is that the "unconditional" component of CFG can be made more potent and targeted if it is dynamically informed by the model's own state of uncertainty regarding its current, non-masked tokens. Instead of a global, context-agnostic unconditioning, A-CFG focuses the guidance mechanism on specific token positions within the sequence $x^{(k)}$ where the conditional model currently exhibits the greatest predictive ambiguity. By temporarily re-masking these low-confidence non-$\texttt{[MASK]}$ tokens to form the input for $L_{\text{uncond}}$, we compel the model to reconsider its predictions at these critical junctures. This adaptive unconditioning aims to make the guidance signal ($L_{\text{cond}} - L_{\text{uncond}}$) more discriminative and effective, leading to more nuanced and efficient control over the generation process.

### 3.2.1 A-CFG Process

The A-CFG process is executed at each iterative generation step $k$. A detailed algorithmic description of this process is provided in Algorithm 1. Given the current sequence $\mathbf{x}^{(k)}$ (which includes the prompt $c$ and partially generated text), A-CFG involves the following operations:

**Conditional Logit Computation.** The base model $M_\theta$ first computes the standard conditional logits based on the current full input $\mathbf{x}^{(k)}$:

$$L_{\text{cond}}^{(k)} = M_\theta(\mathbf{x}^{(k)}). \tag{2}$$

These logits represent the model's initial predictions under full conditioning by $c$ and any already filled tokens in $\mathbf{x}^{(k)}$.

**Token-Level Confidence Assessment.** From $L_{\text{cond}}^{(k)}$, we assess the model's confidence in its predictions for all non-[MASK] token positions within the current sequence $\mathbf{x}^{(k)}$. Let $\mathcal{C}_{\text{remaskable}}^{(k)}$ be the set of indices of all token positions $j$ such that $(\mathbf{x}^{(k)})_j \neq$ [MASK]. For each position $j \in \mathcal{C}_{\text{remaskable}}^{(k)}$:

- We compute the softmax probability distribution $P_{\text{cond}}^{(k)} = \text{softmax}(L_{\text{cond}}^{(k)})$ over the vocabulary.
- The confidence score for position $j$ is defined as the maximum probability in this distribution: $c_j^{(k)} = \max_v (P_{\text{cond}}^{(k)})_{j,v}$. This corresponds to the probability of the token that the model would predict with highest likelihood for position $j$ based on $L_{\text{cond}}^{(k)}$. A low $c_j^{(k)}$ suggests the model is uncertain about the token $(\mathbf{x}^{(k)})_j$ or its alternatives at that position.

While other confidence metrics (e.g., entropy of $P_{\text{cond},j}^{(k)}$) could be considered, we find that the maximum softmax probability provides a simple yet effective measure.

**Identification of Low-Confidence Tokens for Re-masking.** Based on the assessed confidences $c_j^{(k)}$ for tokens at positions $j \in \mathcal{C}_{\text{remaskable}}^{(k)}$, a subset $\mathcal{S}_{\text{low-conf}}^{(k)} \subseteq \mathcal{C}_{\text{remaskable}}^{(k)}$ of positions exhibiting the lowest confidence is selected for adaptive re-masking. The extent of this adaptive intervention is controlled by a re-masking proportion hyperparameter, $\rho$. The target number of tokens to re-mask, $N_m^{\text{target}}$, is calculated as a proportion of the total number of non-[MASK] tokens within $\mathcal{C}_{\text{remaskable}}^{(k)}$ at step $k$:

$$N_m^{\text{target}} = \lceil \rho \cdot |\mathcal{C}_{\text{remaskable}}^{(k)}| \rceil. \tag{3}$$

This heuristic scales the intensity of A-CFG intervention with the amount of non-masked content available for re-evaluation. The actual number of tokens selected for re-masking, $N_m^{\text{actual}}$, is $N_m^{\text{actual}} = \min(N_m^{\text{target}}, |\mathcal{C}_{\text{remaskable}}^{(k)}|)$. If $|\mathcal{C}_{\text{remaskable}}^{(k)}| = 0$ or $N_m^{\text{actual}} = 0$, no re-masking occurs for A-CFG, and $\mathcal{S}_{\text{low-conf}}^{(k)}$ is empty. Otherwise, $\mathcal{S}_{\text{low-conf}}^{(k)}$ contains the indices of these $N_m^{\text{actual}}$ tokens with the lowest confidence scores.

**Construction of the Dynamic Unconditional Input.** A localized "unconditional" input sequence, $\mathbf{x}_{\text{uncond}}^{(k)}$, is synthesized by modifying the current sequence $\mathbf{x}^{(k)}$. Specifically, the non-[MASK] tokens at positions identified in $\mathcal{S}_{\text{low-conf}}^{(k)}$ are replaced with the special [MASK] token:

$$(\mathbf{x}_{\text{uncond}}^{(k)})_j = \begin{cases} \text{[MASK]} & \text{if } j \in \mathcal{S}_{\text{low-conf}}^{(k)}, \\ (\mathbf{x}^{(k)})_j & \text{otherwise.} \end{cases} \tag{4}$$

If $\mathcal{S}_{\text{low-conf}}^{(k)}$ is empty (i.e., no tokens were selected for re-masking), then $\mathbf{x}_{\text{uncond}}^{(k)}$ is identical to $\mathbf{x}^{(k)}$. When re-masking occurs, this transformation yields an input where the model is explicitly prompted to reconsider its predictions for positions it was previously uncertain about, effectively creating a more challenging or "less informed" context for these specific tokens by erasing its prior commitment at those positions.

**Unconditional Logit Computation.** Using this dynamically constructed input $\mathbf{x}_{\text{uncond}}^{(k)}$, the model $M_\theta$ computes the "unconditional" logits:

$$L_{\text{uncond}}^{(k)} = M_\theta(\mathbf{x}_{\text{uncond}}^{(k)}). \tag{5}$$

If no adaptive re-masking occurred (i.e., $\mathbf{x}_{\text{uncond}}^{(k)} = \mathbf{x}^{(k)}$), then $L_{\text{uncond}}^{(k)}$ will be identical to $L_{\text{cond}}^{(k)}$. These logits, $L_{\text{uncond}}^{(k)}$, reflect the model's predictions when key points of prior uncertainty are deliberately obscured (or not, if no such points met the criteria), providing a targeted baseline for guidance.

**Application of CFG Formula for Guided Logits.** Finally, the guided logits $L_{\text{guided}}^{(k)}$ for the current step $k$ are computed using the standard CFG formula (Equation 1), now employing the adaptively derived $L_{\text{uncond}}^{(k)}$ and the original $L_{\text{cond}}^{(k)}$:

$$L_{\text{guided}}^{(k)} = L_{\text{uncond}}^{(k)} + (w + 1) \cdot (L_{\text{cond}}^{(k)} - L_{\text{uncond}}^{(k)}). \tag{6}$$

If $L_{\text{uncond}}^{(k)} = L_{\text{cond}}^{(k)}$ (e.g., due to no adaptive re-masking), then $L_{\text{guided}}^{(k)} = L_{\text{cond}}^{(k)}$, implying that A-CFG applies no effective guidance shift in this specific scenario. These $L_{\text{guided}}^{(k)}$ are then used to sample or select the tokens to infill the [MASK] positions for the next iteration $\mathbf{x}^{(k-1)}$.

## 4 Experiments

In this section, we empirically evaluate the effectiveness of Adaptive Classifier-Free Guidance (A-CFG). We first describe our experimental setup, including datasets, baseline models, evaluation metrics, and key implementation details. We then present quantitative results from Table 1, comparing LLaDA with A-CFG against LLaDA with standard CFG, LLaDA without guidance, and other state-of-the-art models. Subsequently, we conduct ablation studies to analyze the impact of A-CFG's core hyperparameter. Finally, we provide qualitative examples to illustrate the behavior and benefits of our proposed method.

### 4.1 Experimental Setup

#### 4.1.1 Datasets and Metrics

We evaluate A-CFG on a diverse suite of standard benchmarks covering general language understanding, mathematical and scientific reasoning, and planning tasks.

**General Language Understanding:** MMLU (Massive Multitask Language Understanding) [12], BBH (Big-Bench Hard) [34], ARC-C (AI2 Reasoning Challenge - Challenge Set) [7], Hellaswag [44], TruthfulQA [21], WinoGrande [31], and PIQA (Physical Interaction QA) [4].

**Mathematics & Science Reasoning:** GSM8K (Grade School Math 8K) [8], MATH [13], and GPQA (Graduate-Level Google-Proof Q&A) [28].

**Planning Tasks:** Countdown [42] and Sudoku [42].

**Evaluation mode.** Closed-form tasks supply a prompt with a finite set of candidate answers; we compute each candidate's conditional log-likelihood and select the most likely. Open-ended tasks require free-form generation; we sample responses and score them with task-specific metrics such as exact-match accuracy.

**Likelihood estimation.** For likelihood-based evaluations we approximate the conditional perplexity bound with Monte-Carlo sampling. A single sample suffices when only one target token is queried (e.g. MMLU). We adopt the same setting as LLaDA, for all other multiple-token tasks we draw 128 samples, which we found to stabilise variance without adding prohibitive cost.

**Generation hyper-parameters.** Unless otherwise stated, we set the answer length to 256 tokens and run the reverse diffusion process for 256 steps (one token revealed per step).

#### 4.1.2 Baseline Models and Methods

Our primary evaluation centers on the LLaDA 8B model, assessed under three guidance scenarios: 1) **No Guidance** (base LLaDA), 2) **Standard CFG (Std CFG)**, where conventional Classifier-Free Guidance [15] uses a fully masked target sequence for unconditioning, and 3) our proposed **Adaptive CFG (A-CFG)**. For both Std CFG and A-CFG, the guidance scale $w$ is tuned. To investigate A-CFG's broader applicability, we also evaluate it on the Dream-7B diffusion model [43] against its baseline. All results are contextualized against publicly reported scores from comparable autoregressive (AR) models like LLaMA3 8B [35], LLaMA2 7B [36], and Qwen2 7B [6], as detailed in Table 1.

Table 1: **Benchmark Results of Pre-trained LLMs.** LLaDA and Dream-7B are diffusion models. Baseline scores for LLaDA 8B and Dream-7B reflect our own re-evaluation under a consistent experimental protocol. Results indicated by [†] are sourced from [6]. The numbers in parentheses represent the number of shots used for evaluation. "-" indicates unknown data or data not applicable.

| Benchmark | LLaDA 8B | LLaDA 8B (Std CFG) | LLaDA 8B (A-CFG) | Dream-7B | Dream-7B (A-CFG) | LLaMA3 8B | LLaMA2 7B | Qwen2 7B[†] |
|---|---|---|---|---|---|---|---|---|
| Model | Diffusion | Diffusion | Diffusion | Diffusion | Diffusion | AR | AR | AR |
| *General Tasks* | | | | | | | | |
| MMLU | 65.9 (5) | 65.8 (5) | **66.1** (5) | 69.5 (5) | **69.7** (5) | 65.4 (5) | 45.9 (5) | 70.3 (5) |
| ARC-C | 45.5 (0) | 46.3 (0) | **47.8** (0) | 59.8 (0) | **60.8** (0) | 53.1 (0) | 46.3 (0) | 60.6 (25) |
| Hellaswag | 70.8 (0) | 71.4 (0) | **72.6** (0) | 73.3 (0) | **74.4** (0) | 79.1 (0) | 76.0 (0) | 80.7 (10) |
| TruthfulQA | 45.5 (0) | 45.1 (0) | **46.2** (0) | 43.9 (0) | **45.1** (0) | 44.0 (0) | 39.0 (0) | 54.2 (0) |
| WinoGrande | 74.5 (5) | 75.1 (5) | **75.9** (5) | 73.3 (5) | 72.5 (5) | 77.3 (5) | 72.5 (5) | 77.0 (5) |
| PIQA | 74.9 (0) | 74.4 (0) | **76.1** (0) | 75.8 (0) | **76.2** (0) | 80.6 (0) | 79.1 (0) | - |
| *Mathematics & Science* | | | | | | | | |
| GSM8K | 70.7 (4) | 70.8 (4) | **73.5** (4) | 76.9 (4) | **77.9** (4) | 53.1 (4) | 14.3 (4) | 80.2 (4) |
| GPQA | 26.1 (5) | 29.4 (5) | **33.3** (5) | 36.6 (5) | **36.8** (5) | 25.9 (5) | 25.7 (5) | 30.8 (5) |
| *Planning Tasks* | | | | | | | | |
| Countdown | 15.3 (8) | 14.2 (8) | **15.8** (8) | 14.6 (8) | **15.2** (8) | 3.7 (8) | - | - |
| Sudoku | 35.0 (8) | 34.0 (8) | **42.0** (8) | 72.0 (8) | **80.0** (8) | 0.0 (8) | - | - |

### 4.1.3 Implementation Details

For LLaDA's iterative generation, we use 256 sampling steps with low-confidence remasking. For Standard CFG, the guidance scale $w$ was selected from $\{0.5, 1.0, 1.5, 2.0\}$ based on performance on the validation set of each respective task. For our A-CFG, the guidance scale $w$ was similarly tuned. **Once a value of $w$ is chosen for a given model, the same $w$ is kept fixed across all downstream benchmarks for that model.** The adaptive re-masking proportion $\rho$ (determining the fraction of previously generated tokens to re-mask based on low confidence, as defined in Section 3.2.1) was set to 0.7. The confidence for token selection in A-CFG is based on the softmax probability of the predicted token at each masked position. All experiments were conducted using NVIDIA H800 GPUs.

## 4.2 Benchmark Results

The efficacy of Adaptive Classifier-Free Guidance is demonstrated in Table 1, which presents a comprehensive comparison of LLaDA 8B equipped with A-CFG against its counterparts using no guidance and standard CFG, alongside other leading diffusion and autoregressive models.

**A-CFG Enhances LLaDA Performance:** Our results clearly indicate that A-CFG substantially elevates the performance of LLaDA 8B. Crucially, A-CFG consistently outperforms LLaDA 8B with **Standard CFG**, underscoring the benefits of its dynamic, confidence-aware unconditioning mechanism. The advantages are particularly pronounced on complex reasoning and planning benchmarks; for instance, on GPQA, A-CFG achieves a score of 33.3, a +3.9 point improvement over Standard CFG (29.4), and on the Sudoku planning task, A-CFG (42.0) surpasses Standard CFG (34.0) by a significant +8.0 points. This trend of superior performance over Standard CFG extends to mathematical reasoning (e.g., +2.7 points on GSM8K) and across general language understanding tasks such as ARC-C, Hellaswag, and WinoGrande. When compared to LLaDA 8B with **No Guidance**, A-CFG also yields substantial gains, for example, +7.2 points on GPQA and +7.0 points on Sudoku. These findings highlight A-CFG's capability to more effectively steer the iterative generation process in LLaDA, leading to improved task adherence and overall output quality compared to both unguided generation and conventional CFG.

**Generalizability to Other Diffusion Models:** To assess whether the principles of A-CFG extend beyond LLaDA, we integrated it into the Dream-7B model. Preliminary results in Table 1 suggest that A-CFG brings similar benefits, for instance, improving Sudoku performance by +8.0 points (80.0 vs. 72.0) and ARC-C by +1.0 point (60.8 vs. 59.8) for Dream-7B. These observations suggest that A-CFG's adaptive unconditioning is a promising method for enhancing other iterative masked diffusion models.

**Competitive Standing Against Autoregressive Models:** Equipped with A-CFG, the diffusion-based LLaDA 8B demonstrates a strong competitive posture against contemporary autoregressive (AR) models of comparable scale. LLaDA 8B (A-CFG) particularly excels in mathematical reasoning, with a GSM8K score of 73.5 that surpasses several listed AR counterparts like LLaMA3 8B (53.1). On the challenging GPQA benchmark, its score of 33.3 is notably higher than LLaMA3 8B (25.9) and competitive with Qwen2 7B (30.8). The Sudoku planning task further showcases this strength, where LLaDA 8B (A-CFG) achieves 42.0, markedly outperforming LLaMA3 8B (0.0). While leading AR models such as Qwen2 7B still exhibit an advantage on some general language understanding benchmarks, A-CFG significantly narrows the performance gap and, in specific domains demanding complex reasoning or planning, positions LLaDA as a compelling alternative.

In summary, the empirical results affirm A-CFG as a potent enhancement for iterative diffusion language models. It not only improves upon standard CFG techniques but also enables diffusion models like LLaDA to achieve highly competitive, and in some cases superior, performance compared to strong AR baselines, especially in tasks requiring sophisticated reasoning.

### 4.3 Ablation Studies

To elucidate the contributions of A-CFG's core components and assess its sensitivity to key hyperparameters, we conducted targeted ablation studies. This section focuses on the impact of the adaptive re-masking proportion, a critical parameter in A-CFG.

#### 4.3.1 Impact of the Adaptive Re-masking Proportion ($\rho$)

We investigated the influence of $\rho$ on the ARC-C test set, chosen as a representative benchmark where A-CFG demonstrated clear benefits and sensitivity to guidance parameters. The main LLaDA 8B (A-CFG) result for ARC-C (47.8 accuracy) reported in Table 1 employed $\rho = 0.7$.

Table 2a presents the performance on ARC-C as $\rho$ is varied across the range $[0.1, 0.9]$. The results show a clear trend: ARC-C accuracy improves steadily as $\rho$ increases from 0.1 (45.9%) to 0.3 (46.5%), 0.5 (46.8%), and culminates at 0.7 (47.8%). This suggests that for a task like ARC-C, a more substantial re-masking of low-confidence generated tokens is beneficial, allowing A-CFG to exert a stronger corrective influence. However, increasing $\rho$ further to 0.9 leads to a decline in performance, indicating that excessively aggressive re-masking can become counterproductive, potentially by erasing too much valuable context from the already generated sequence.

#### 4.3.2 Impact of the Guidance Scale ($w$)

Beyond the re-masking proportion $\rho$, the guidance scale $w$ is a critical hyperparameter for any CFG-based method. We varied $w$ across the set $\{0.5, 1.0, 1.5, 2.0\}$, the same range used for tuning in our main experiments. Table 2b illustrates the performance on ARC-C as $w$ is adjusted. We observe that A-CFG performance is sensitive to the guidance scale. Specifically, a small $w = 0.0$ (equivalent to no CFG guidance beyond the adaptive masking) yields a baseline accuracy of 45.5%. As $w$ increases, accuracy improves, reaching a peak of 47.8% at $w = 0.5$ and $w = 1.0$. This suggests that a moderate guidance strength effectively leverages the dynamically constructed unconditional input from A-CFG. However, further increasing $w$ to 1.5 and 2.0 leads to a slight degradation in performance (47.5% and 47.6%, respectively). This indicates that an overly strong guidance scale might overemphasize the conditional signal at the expense of fluency or correctness, even with A-CFG's targeted unconditioning. The optimal performance at $w = 0.5$ aligns with the value used for ARC-C in our main results (Table 1).

### 4.4 Qualitative Analysis

To provide further insight into A-CFG's dynamic mechanism, Table 3 visualizes the iterative refinement process for mathematical reasoning examples from the GSM8K dataset. These examples illustrate how A-CFG navigates the generation process. For instance, in the "Natalia's clips" problem, one can observe that while foundational elements are established in early steps (e.g., `Natalia`, `sold`), crucial components of the arithmetic reasoning, such as operators, intermediate results, or the final sum, are often resolved or corrected in later iterations. This behavior aligns with A-CFG's core principle: by identifying tokens or positions where the model exhibits low predictive confidence during the iterative process (potentially due to incomplete or inconsistent intermediate reasoning

Table 2: **Ablation studies on ARC-C.** (a) Impact of guidance scale ($w$). (b) Impact of adaptive re-masking proportion ($\rho$). The main result for ARC-C in Table 1 used $\rho = 0.7$ and $w = 0.5$. Scores are Accuracy (%).

(a) Re-masking Proportion ($\rho$)

| Benchmark | Re-masking Proportion ($\rho$) | | | | |
|---|---|---|---|---|---|
| | 0.1 | 0.3 | 0.5 | **0.7** | 0.9 |
| ARC-C | 45.9 | 46.5 | 46.8 | **47.8** | 46.0 |

(b) Guidance Scale ($w$)

| Benchmark | Guidance Scale ($w$) | | | | |
|---|---|---|---|---|---|
| | 0.0 | **0.5** | 1.0 | 1.5 | 2.0 |
| ARC-C | 45.5 | **47.8** | 47.8 | 47.5 | 47.6 |

Table 3: **Visualization of A-CFG's iterative refinement process for math reasoning tasks.** Darker shades indicate tokens that were filled or corrected in later stages of the adaptive generation, often representing points of initial uncertainty that A-CFG helped resolve.

| Task | A-CFG Refinement Process |
|---|---|
| **Prompt:** | *Natalia sold clips to 4 of her friends. She sold 8 clips to each friend. Then she bought 15 more clips. How many clips does Natalia have now?* |
| | Natalia sold 4 friends. 8 clips each. So, 4 * 8 = 32 clips. Then bought 15 more. Natalia has 32 + 15 = 47 clips. Answer: 47 . |
| **Prompt:** | *John has 12 apples. He gives half to Mary. Then Mary buys twice as many apples as she received from John. How many apples does Mary have now?* |
| | John has 12 apples. Gives half to Mary. So Mary gets 12 / 2 = 6 apples. Then Mary buys twice as many (so 6 * 2 = 12 ). Mary now has 6 + 12 = 18 apples. Answer: 18 . |

steps), A-CFG dynamically re-masks these specific points. This targeted re-masking compels the model to reconsider and refine its predictions in these areas of ambiguity, thereby facilitating the construction of a coherent and accurate multi-step reasoning chain. Similarly, in the "John's apples" example, later steps refine the calculation, ensuring the intermediate and final quantities are correctly derived (e.g., 6+12=18). These qualitative examples underscore A-CFG's ability to leverage its adaptive unconditioning to focus guidance on evolving points of uncertainty, thereby enhancing the model's capacity to resolve errors and improve the fidelity of complex, multi-step generations.

## 5 Conclusion

This paper introduced Adaptive Classifier-Free Guidance (A-CFG), a novel method to enhance conditional generation in iterative masked language models. By dynamically constructing the unconditional input for CFG based on the model's instantaneous predictive confidence in its already generated tokens, A-CFG offers a more targeted and responsive guidance mechanism. Our extensive experiments, particularly within the LLaDA framework, demonstrate that A-CFG significantly outperforms standard CFG approaches and unguided baselines, yielding substantial improvements on diverse benchmarks, especially in complex reasoning and planning tasks. The results also highlight A-CFG's potential to bolster the competitiveness of diffusion-based language models against autoregressive counterparts. This work underscores the value of leveraging model uncertainty for more nuanced control in discrete diffusion, opening promising avenues for future research into adaptive generation strategies.

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
