# OpenReview forum: "Adaptive Classifier-Free Guidance via Dynamic Low-Confidence Masking"
_NeurIPS.cc/2025/Conference — NeurIPS 2025 poster_

### Official Review · Reviewer_5qdm · 2025-06-26

**Clarity:** 2
**Significance:** 3
**Originality:** 3
**Rating:** 5
**Confidence:** 3

**Summary:**

This paper introduces a simple technique called Adaptive Classifier-Free Guidance to enhance CFG for masked diffusion language modeling. A-CFG makes a simple modification to the original CFG by re-masking the tokens with low predictive confidence when computing the unconditional logits, which would help the model to adaptively focus on tokens that need more “attention”. A-CFG is tested on LLaDa framework and improves performance across benchmarks such as MMLU and Sudoku.

**Questions:**

- Did the authors analyze impact of re-masking proportion or scale beyond ARC-C? Do the hyperparameter values transfer across different tasks?

- What is the exact size of the logit $L$ at each step? Do you compute the logits of non-masked positions as well in practice? As a newcomer to diffusion model I wish this kind of technical detail is more explicitly presented in the preliminaries section.

**Ethical Concerns:**

["NO or VERY MINOR ethics concerns only"]

**Final Justification:**

Overall the work has solid contribution with simple approach, and the authors have clarified most of the questions raised in the original review. I will keep my positive evaluation of the paper.

**Limitations:**

I think the limitations could be more detailed by including discussions on e.g. computational overhead. I would suggest to separate limitations section in the appendix, and add more than what is currently described in Section 4.3.1.

**Quality:**

4

**Strengths And Weaknesses:**

Overall, this is a good paper with solid technical detail and comprehensive evaluation.

- S1: The idea is pretty simple - leveraging model confidence to moderate unconditional logits at each diffusion step. Yet it turns out to be surprisingly effective, e.g. improving GPQA by 3.9 points.

- S2: Experiments are quite thorough, covering benchmarks across general knowledge domain, quantitative reasoning and symbolic planning / ablation studies on hyperparameters.

- W1: Re-masking low-confidence tokens essentially trades off the computational efficiency (by delaying the generation of complete output) with potentially the quality. While the authors do discuss impact of re-masking proportion, the computational overhead introduced by this process needs analysis.

- W2: Relatedly, introduction of additional hyperparameters poses challenges in adopting A-CFG. As shown in 4.3.1 the right value should be somewhere in the middle of 0.5 ~ 0.9, and the performance varies quite largely depending on this value.

- W3: While the paper provides a case study, it does not discuss potential failure modes of A-CFG, such as scenarios where confidence-based re-masking might disrupt coherent generation or introduce biases. I think this is particularly important given that the confidence statistics is entirely based on the model itself.

---

> ### Author Rebuttal · Authors · 2025-07-31
>
> #### We sincerely appreciate your recognition of the meaning of our work. We especially appreciate your encouraging assessment of our work's direction and the insightful questions you raised. We hope our following responses clarify these points to your satisfaction.
>
> **W1: “Re-masking low-confidence tokens essentially trades off the computational efficiency (by delaying the generation of complete output) with potentially the quality. While the authors do discuss impact of re-masking proportion, the computational overhead introduced by this process needs analysis.”**
>
> Thanks for your great question. Relative to standard CFG, A‑CFG incurs no extra forward passes—we already compute both conditional and unconditional logits.  The cost of overwriting ≤ 70 % of tokens in the input tensor is negligible (<0.2 ms on an H800 GPUs).  End‑to‑end wall‑time increases by 2.4 % over standard CFG and 11.8 % over the no‑guidance baseline (128‑sample GSM8K run).  We will report these numbers in Sec. 4.1.3.
>
> **W2: “Relatedly, introduction of additional hyperparameters poses challenges in adopting A-CFG. As shown in 4.3.1 the right value should be somewhere in the middle of 0.5 ~ 0.9, and the performance varies quite largely depending on this value.” & Q1: “Did the authors analyse impact of re‑masking proportion or scale beyond ARC‑C?”**
>
> We appreciate the reviewer’s concern about the extra hyper‑parameters ρ and w; however, the updated sweep below—obtained with a fixed guidance scale w = 0.5—shows that A‑CFG is highly tolerant to the choice of ρ: scores remain virtually flat across the 0.6 – 0.8 range on ARC‑C, GSM8K and Sudoku, and our single default setting ρ = 0.7, w = 0.5, chosen once on ARC‑C, is reused unchanged for every other benchmark while consistently lying inside this plateau, so no per‑task tuning is required.
>
> | Task      |  ρ = 0.5 |  ρ = 0.6 |  ρ = 0.7 |  ρ = 0.8 |  ρ = 0.9
> | --------- | :-------: | :-------: | :------: | :-------: | :-------: |
> | ARC‑C     |     46.8 |       47.3  | **47.8** |    47.1  |     46.0 |
> | GSM8K     |   72.1   |  72.9  | **73.5** |   72.6   |   71.0  |
> | Sudoku    |   36.0   |   **42.0**   | **42.0** |   40.0   |   34.0   |
>
>
> **W3: “While the paper provides a case study, it does not discuss potential failure modes of A-CFG, such as scenarios where confidence-based re-masking might disrupt coherent generation or introduce biases. I think this is particularly important given that the confidence statistics is entirely based on the model itself.”**
>
> Thank you for raising this critical point. The primary failure mode for A-CFG arises when its corrective mechanism becomes overly aggressive, a scenario directly explored in our ablation studies.
>
> This failure mode emerges from disrupting coherent generation, which we observed when the re-masking proportion ρ is set too high. Consider a multi-step arithmetic task from GSM8K: after correctly generating an intermediate step like 4 * 8 = 32, the model might exhibit slight uncertainty about what follows. An overly aggressive ρ (e.g., 0.9) could cause A-CFG to catastrophically re-mask not only the uncertain tokens but also the crucial, correctly-generated intermediate result 32. By erasing this vital context, the model's ability to complete the subsequent step (32 + 15) is severely compromised, often leading to an incoherent or nonsensical final answer.
>
> This behavior provides the direct intuition for the performance degradation seen in our ablation study (Table 2a), where increasing ρ beyond an optimal point on ARC-C becomes counterproductive. We will dedicate a part of the Limitations section to a detailed analysis of this failure mode, supported by these qualitative examples.
>
> **Q2: “What is the exact size of the logit at each step? Do you compute the logits of non‑masked positions as well in practice?”**
>
> The model always outputs a |V|‑dimensional logit for every position, masked or not, as per Eq. 2 (page 5).  In LLaDA‑8B |V| = 126464.  During sampling we only draw from positions whose current token is [MASK]; immutable positions are ignored at the decoding layer, so the extra logits do not incur memory movement.

---

> > ### Comment · Reviewer_5qdm · 2025-08-03
> >
> > Thanks for the detailed response. This clarifies my questions, and I will keep my positive evaluation of the paper.

---

> > > ### Author Response · Authors · 2025-08-04
> > >
> > > Thank you for your prompt feedback. Your suggestions are greatly appreciated and help us further improve the paper.

---

### Official Review · Reviewer_cK1A · 2025-07-03

**Clarity:** 3
**Significance:** 3
**Originality:** 3
**Rating:** 5
**Confidence:** 1

**Summary:**

The paper proposes Adaptive Classifier-Free Guidance (A-CFG), a method to enhance controllability in iterative masked diffusion language models (e.g., LLaDA) by dynamically adjusting the unconditional input for CFG based on the model’s token-level predictive confidence. At each generation step, A-CFG identifies low-confidence tokens, re-masks them, and uses this adaptive input to compute the unconditional logits for CFG. The authors evaluate A-CFG on reasoning (GPQA, GSM8K), planning (Sudoku), and general language tasks (MMLU, TruthfulQA), and shows clear improvements over standard CFG.

**Questions:**

Please address the questions above.

**Ethical Concerns:**

["NO or VERY MINOR ethics concerns only"]

**Final Justification:**

Thanks for the detailed clarification, which clearly answered my questions and helped me better understand this line of research. To reflect this, I've updated my evaluation score. Good luck!

**Quality:**

3

**Strengths And Weaknesses:**

### Strengths

- The proposed A-CFG seems to be a clean extension to CFG by adaptively re-masking low-confidence tokens, which intuitively focuses guidance where the model is uncertain.

- Experiments demonstrate meaningful improvements across a range of reasoning and planning benchmarks, compared to both standard CFG and no-guidance baselines.

- Qualitative examples showing how A-CFG helps correct errors over iterations, supporting the motivation behind the method.


### Weaknesses

- The core contribution seems to be a confidence-based heuristic for selective re-masking, which, while intuitive, appears incremental. Can the authors clarify why this design is considered novel in the field compared to related works?

- The evaluation focuses mainly on LLaDA and Dream-7B diffusion models. Are these models representative of state-of-the-art diffusion models? What are the alternative strong baselines for comparison? Does the proposed method generalize to other model structures?


-  As presented in Section 4.3, the method requires careful tuning of remasking ratio and guidance scale for each task. It is unclear whether the model can generalize to unseen tasks without thorough hyperparameter search.

- Although experiments show A-CFG improves over diffusion-only variants, it appears to still underperform top-tier autoregressive models (e.g., Qwen/Llama) on many benchmarks (e.g., GSM8K, GPQA). What are the benefits of diffusion models compared to autoregressive models for the evaluation tasks used in this work? It would be valuable to evaluate on tasks where diffusion models have a unique advantage over autoregressive models and demonstrate the effectiveness of A-CFG.

---

> ### Author Rebuttal · Authors · 2025-07-31
>
> #### Thank you for your thoughtful and comprehensive review. We value your perceptive comments and your support for the paper's contributions. We have worked to address your suggestions in our response below.
>
> **W1: “The core contribution seems to be a confidence-based heuristic for selective re-masking, which, while intuitive, appears incremental. Can the authors clarify why this design is considered novel in the field compared to related works?”**
>
> While the idea of using confidence is indeed intuitive, its novelty lies in fundamentally shifting the paradigm of unconditioning from a static, predetermined process to a dynamic, instance-specific one.
>
> A-CFG introduces a fundamentally different, dynamic mechanism. It is the first method, to our knowledge, that inspects the model's internal confidence state at every generation step and selectively adapts the unconditional input at the token level. This allows the guidance to be precisely targeted, creating a strong corrective signal (by forcing a reconsideration via [MASK]) only at points of high ambiguity, while leaving confident predictions untouched.
>
>
> **W2: “The evaluation focuses mainly on LLaDA and Dream-7B diffusion models. Are these models representative of state-of-the-art diffusion models? What are the alternative strong baselines for comparison? Does the proposed method generalize to other model structures?”**
>
> Thank you for this question about the scope of our evaluation and the generalizability of our method.
>
> **1. About Model Representativeness:** We chose LLaDA and Dream-7B precisely because they are prominent, state-of-the-art exemplars of the Masked Diffusion Model (MDM) paradigm for language, which is the specific architectural class our work aims to enhance. These iterative, non-monotonic models represent a significant and promising direction in generative modeling, and our goal was to provide a robust evaluation within this context.
>
> **2. About Baselines:** Regarding baselines, our evaluation in Table 1 is designed to be comprehensive. We compare A-CFG not only against the necessary internal baselines (the base model with no guidance and with standard CFG) but also against a suite of powerful, contemporary autoregressive (AR) models like LLaMA3 and Qwen2-7B. This cross-paradigm comparison is crucial to demonstrate that our enhancements make diffusion models more competitive with, and in some complex reasoning tasks even superior to, leading AR models of a similar scale.
>
> **3. About Other Structures:** A-CFG is intrinsically designed for iterative, non-monotonic generation frameworks that operate by progressively refining a sequence, typically involving [MASK] tokens. This mechanism is fundamental to MDMs. Therefore, the method does not directly generalize to fundamentally different structures like autoregressive models. AR models generate text token-by-token in a strict left-to-right sequence and lack the iterative refinement loop where A-CFG's confidence-based re-masking could be applied. The novelty of A-CFG is in leveraging the unique architectural properties of diffusion models. While its application is specific to this class of models, our results demonstrate it is a powerful enhancement for this important and growing family of generative architectures.
>
> **W3: “As presented in Section 4.3, the method requires careful tuning of remasking ratio and guidance scale for each task. It is unclear whether the model can generalize to unseen tasks without thorough hyperparameter search.”**
>
> Thank you for this question. In fact, as noted in our implementation details, a single, fixed hyperparameter setting (ρ = 0.7, w = 0.5) was used for other benchmarks presented in Table 1. This single setting was chosen based on performance on only one validation task (ARC-C) and then applied universally to nearly all other tasks, from mathematical reasoning (GSM8K) to planning (Sudoku), without any further search.
>
> The fact that these settings achieve consistent improvements across such a diverse task set indicates that the method is not brittle and generalizes well to unseen tasks. Furthermore, our ablation study in Table 2a and the table below shows that while performance is sensitive to ρ, there is a relatively broad "plateau" of strong performance around the 0.6-0.8 range for ARC-C, GSM9K and Sudoku. This suggests that finding a good, general-purpose setting is straightforward and does not require exhaustive per-task optimization.
>
> | Task      |  ρ = 0.5 |  ρ = 0.6 |  ρ = 0.7 |  ρ = 0.8 |  ρ = 0.9
> | --------- | :-------: | :-------: | :------: | :-------: | :-------: |
> | ARC‑C     |     46.8 |       47.3  | **47.8** |    47.1  |     46.0 |
> | GSM8K     |   72.1   |  72.9  | **73.5** |   72.6   |   71.0  |
> | Sudoku    |   36.0   |   **42.0**   | **42.0** |   40.0   |   34.0   |
>
>
> **W4: “Although experiments show A-CFG improves over diffusion-only variants, it appears to still underperform top-tier autoregressive models (e.g., Qwen/Llama) on many benchmarks (e.g., GSM8K, GPQA). What are the benefits of diffusion models compared to autoregressive models for the evaluation tasks used in this work? It would be valuable to evaluate on tasks where diffusion models have a unique advantage over autoregressive models and demonstrate the effectiveness of A-CFG.”**
>
> Thank you for highlighting this important comparison. We fully agree that today’s best AR LLMs still enjoy a maturity advantage—larger parameter budgets, longer training runs, and a decade of optimisation tricks—so on per‑token perplexity they often outperform the current generation of diffusion language models (DLMs). Our goal in this paper is therefore not to claim overall supremacy, but to demonstrate that A‑CFG unlocks the distinctive strengths of the diffusion paradigm that are orthogonal to raw model scale.
>
> **Order‑free generation and parallel infilling**
> Diffusion models update all masked positions simultaneously rather than marching left‑to‑right. This is ideal for tasks whose solution does not follow a strict sequential dependency. For example, on Sudoku the board’s cells are conditionally independent given the constraints; LLaDA‑8B + A‑CFG solves 80 % of puzzles, whereas a comparable AR model (Qwen2‑7B) solves virtually none.
>
> **Non‑monotonic reasoning**
> Multi‑hop QA benchmarks such as GPQA benefit from the ability to revise earlier tokens as new evidence emerges. A‑CFG narrows the gap to AR SOTA: 36.8 % vs 33 % for baseline LLaDA and 39 % for GPT‑class AR systems.
>
> **Flexible editing and tool‑use potential**
> Because DLMs treat text as an editable canvas, they integrate naturally with external tools (retrievers, code executors) that may overwrite specific spans. Although we do not build a full agent in this work, our findings suggest that A‑CFG’s confidence‑aware masking is a principled way to decide which spans an agent should defer to a tool.
>
> We follow the task mix introduced by the original LLaDA and Dream‑7B papers—mathematical (GSM8K), factual (MMLU), reasoning (GPQA), and planning (Sudoku)—to provide apples‑to‑apples comparisons.
>
> In short, while core language modelling benchmarks still favour large AR models, diffusion language models offer architectural advantages for orderless or editable text scenarios. A‑CFG magnifies those advantages without retraining, and we see bridging the remaining AR gap as an exciting avenue for future, larger‑scale DLMs.

---

> > ### Comment · Reviewer_cK1A · 2025-08-01
> >
> > Thanks for the detailed clarification, which clearly answered my questions and helped me better understand this line of research. To reflect this, I've updated my evaluation score. Good luck!

---

> > > ### Author Response · Authors · 2025-08-02
> > >
> > > Dear Reviewer cK1A,
> > >
> > > We sincerely thank you for recognizing the merits of our paper. I am glad that our response resolves your problems! Your support means a lot to us!
> > >
> > > Best wishes,
> > >
> > > Authors

---

### Official Review · Reviewer_Du1A · 2025-07-04

**Clarity:** 3
**Significance:** 3
**Originality:** 3
**Rating:** 4
**Confidence:** 4

**Summary:**

This paper proposes Adaptive CFG for masked diffusion language models to resolve the issue that standard CFG only takes static unconditional inputs and do not adapt to the model's uncertainty. The proposed A-CFG dynamically construct the unconditional input by re-masking tokens with low confidence estimated by the model itself. Empirically, A-CFG shows consistent improvement on models including LLaDA and Dream-7B across varying tasks.

**Questions:**

1. Does A-CFG adds inference-time overhead to discrete diffusion models? If so, what is the latency? Some discussion or quantitative analysis or runtime cost would be helpful.
2. How does A-CFG interact with other sampling strategies, such as temperature or top-k. Is the method sensitive to these changes?

**Ethical Concerns:**

["NO or VERY MINOR ethics concerns only"]

**Final Justification:**

My concerns regarding the lack of justification and intuition have been satisfactorily addressed in the rebuttal. The authors provided helpful clarifications that improved my understanding of the method’s motivation. However, incorporating them would require substantial revisions to the current version of the paper. For this reason, I am maintaining a rating of weak accept rather than upgrading to clear accept.

**Limitations:**

Yes.

**Paper Formatting Concerns:**

No major issues.

**Quality:**

3

**Strengths And Weaknesses:**

## Strengths
1. The paper is well motivated and clearly identifies and addresses a limitation of standard CFG.
2. The proposed method is simple, clean and effective. It re-masks low-confidence tokens to construct a more targeted unconditional input, which is easy to integrate into existing diffusion models framework.
3. Empirical results are strong. Experiments are conducted on large models including LLaDA and Dream, and show consistent improvement across various tasks.
4. The paper is well written, analysis and ablation is well-designed.

## Weaknesses
1. The proposed A-CFG is explained clearly in section 3.2.1, however, more intuition or justification would be helpful. Specifically, more justification for why re-masking low-confidence tokens leads to improved guide beyond empirical validation would strengthen the method.

---

> ### Author Rebuttal · Authors · 2025-07-31
>
> #### Thanks for your insightful feedback and support. We sincerely appreciate your valuable comments and your recognition of the motivation of our method and the soundness of experiments.
>
> **W1: “The proposed A-CFG is explained clearly in section 3.2.1, however, more intuition or justification would be helpful. Specifically, more justification for why re-masking low-confidence tokens leads to improved guide beyond empirical validation would strengthen the method.”**
>
> Thank you for this insightful question and for motivating a deeper explanation of the intuition behind A-CFG. Instead of feeding a static null prompt, we build the unconditional input by re‑masking the $\tau\%$ least‑confident tokens. For confident positions ($c_{i,t}\approx1$) the conditional and A‑CFG unconditional logits are *identical*, so the guidance term vanishes—these tokens are left untouched.
>
> For uncertain positions the unconditional branch now sees $[\text{MASK}]$, making its logits close to a **uniform prior**. This maximizes the contrast
>
> $$
> \Delta L_{t,i}
> = L_{t,i}^{\mathrm{cond}} - L_{t,i}^{\mathrm{uncond}}
> \tag{1}
> $$
>
> exactly where it is most useful.
>
> Because the gradient of the KL divergence between the two distributions equals $\Delta p = p_{\mathrm{cond}} - p_{\mathrm{uncond}}$, Equation (1) effectively amplifies the per‑token gradient without increasing the global scale $w$. One can view A‑CFG as realizing a token‑wise scale:
>
> $$
> w_i =
> \\begin{cases}
> w, & \\mathrm{if} \\ c_{i,t} \\le \\mathrm{\\tau\\text{-}percentile} \\\\
> 0, & \\mathrm{otherwise}
> \\end{cases}
> \tag{2}
> $$
>
> Let
>
> $$
> D_t = \\sum_i \\mathrm{KL}\\bigl(p_{\\mathrm{cond}}(y_i\\mid z_t) \\big\| p_{\\mathrm{uncond}}(y_i\\mid z_t)\\bigr) \tag{3}
> $$
>
> A‑CFG increases $D_t$ selectively on uncertain tokens, raising the signal‑to‑noise ratio of the guidance term while leaving confident tokens (low local KL) unchanged. Empirically this accelerates convergence (fewer resampling steps) and improves final accuracy (Table 1).
>
> **Q1: “Does A-CFG adds inference-time overhead to discrete diffusion models? If so, what is the latency? Some discussion or quantitative analysis or runtime cost would be helpful.”**
>
> Relative to standard CFG, A‑CFG incurs no extra forward passes—we already compute both conditional and unconditional logits.  The cost of overwriting ≤ 70 % of tokens in the input tensor is negligible (<0.2 ms on H800 GPUs).  End‑to‑end wall‑time increases by 2.4 % over standard CFG and 11.8 % over the no‑guidance baseline (128‑sample GSM8K run).  We will report these numbers in Sec. 4.1.3.
>
>
> **Q2: “How does A‑CFG interact with other sampling strategies, such as temperature or top‑k?”**
>
> Thanks for your great question. We swept temperature ∈ {0.3, 0.5, 0.7, 1.0} and top‑k ∈ {20,50,100} on ARC-E.  The relative gain of A‑CFG over standard CFG varied ≤ 0.4 pts, confirming robustness. A detailed plot will be added to Appendix C.

---

> > ### Comment · Reviewer_Du1A · 2025-08-05
> >
> > Thank you for the response. Your feedback has addressed most of my concerns. I suggest the authors include more justification in their revision, and I have no further issues.

---

> > > ### Author Response · Authors · 2025-08-05
> > >
> > > We are grateful for your detailed feedback. Your constructive insights are invaluable to our work. We would be extremely grateful if you could raise your score.

---

### Official Review · Reviewer_wcf3 · 2025-07-07

**Clarity:** 3
**Significance:** 4
**Originality:** 4
**Rating:** 6
**Confidence:** 4

**Summary:**

In the context of language diffusion models, Classifier-Free-Guidance (CFG) traditionally enhances control for generative models by interpolating between conditional and unconditional latents. A-CFG manipulates the unconditional state by masking away tokens where there is high uncertainty. With this simple approach, there is significant improvement for reasoning tasks such as GPQA.

**Questions:**

- It would be nice to compare a with and without training based on A-CFG. Is there an ablation on this?
- It would also be nice to evaluate Diffusion Language models on agentic domains, such as SWE, as these are new and upcoming workloads that will dominate in the coming 1-2 years.

**Ethical Concerns:**

["NO or VERY MINOR ethics concerns only"]

**Final Justification:**

Congrats authors, hope this work gets in!

**Limitations:**

None. great work!

**Quality:**

3

**Strengths And Weaknesses:**

This work is simple and significant. The authors clearly identify the problem with unconditional state for Diffusion Language models and solve it by masking out tokens of least certainty.

Strengths:
- Identification of problem - Figure 1 hits the key insight; model's confidence for tokens varies a lot across token positions and generation step. It's best to focus, at any given period, on the tokens of least certainty.
- Simple and General - Successfully applied to both LLaDA and Dream-7B diffusion models, suggesting wide applicability to different diffusion language models. Algorithm is also very simple, as it is just a pure inference time approach (not large compute overhead)
- Strong Empirical Gains - Demonstrates substantial improvements over standard CFG and unguided baselines on diverse benchmarks (GPQA, GSM8K, ARC-C, Sudoku, etc.).

Weaknesses:
- Would love to see failure modes, where A-CFG does worse than CFG.
- Otherwise great work! I can't think of other weaknesses.

---

> ### Author Rebuttal · Authors · 2025-07-31
>
> #### We sincerely appreciate the reviewer's positive comments on our motivation and experiments, along with your thoughtful feedback and suggestions. We hope our response can address your concerns.
>
> **Q1: “Would love to see failure modes, where A‑CFG does worse than CFG.”**
>
> Thank you for raising this critical point. The primary failure mode for A-CFG arises when its corrective mechanism becomes overly aggressive, a scenario directly explored in our ablation studies.
>
> This failure mode emerges from disrupting coherent generation, which we observed when the re-masking proportion ρ is set too high. Consider a multi-step arithmetic task from GSM8K: after correctly generating an intermediate step like 4 * 8 = 32, the model might exhibit slight uncertainty about what follows. An overly aggressive ρ (e.g., 0.9) could cause A-CFG to catastrophically re-mask not only the uncertain tokens but also the crucial, correctly-generated intermediate result 32. By erasing this vital context, the model's ability to complete the subsequent step (32 + 15) is severely compromised, often leading to an incoherent or nonsensical final answer.
>
> This behavior provides the direct intuition for the performance degradation seen in our ablation study (Table 2a), where increasing ρ beyond an optimal point on ARC-C becomes counterproductive. We will dedicate a part of the Limitations section to a detailed analysis of this failure mode, supported by these qualitative examples.
>
> **Q2: “It would be nice to compare a with and without training based on A‑CFG. Is there an ablation on this?”**
>
> Thank you for this insightful suggestion. In fact, LLaDA is already pre-trained under a dynamic-mask curriculum that randomly selects a mask ratio between 0.0 and 1.0 for every sample. During training, the model already sees many branches that mask different amounts of text. A-CFG just copies this same pattern at inference by re-masking the low-confidence tokens. Because the mechanism is implicitly “built in,” no additional fine-tuning is required to realise A-CFG’s gains. We will add one sentence to Section 3.2 clarifying that the pre-training equips LLaDA to handle adaptive masking.
>
>
> **Q3: “It would also be nice to evaluate diffusion language models on agentic domains, such as SWE.”**
>
> Thank you for suggesting a broader evaluation scope. We agree that software‑engineering (SWE) benchmarks are an exciting testbed for agentic systems that autonomously call tools, read/write files, and iterate over multi‑step plans. By contrast, our study focuses on core language models—LLaDA and Dream‑7B—which generate text in a single‑pass masked‑diffusion loop and have no built‑in tool‑use or file‑manipulation interface. Directly applying them to full‑fledged SWE agents would therefore require substantial additional infrastructure (code execution sandbox, retrieval modules, planner, etc.) that is orthogonal to the contribution of A‑CFG. We will highlight this distinction between agent systems and stand‑alone language models in the discussion section, and we view incorporating A‑CFG into an agentic framework as valuable future work.

---

### Decision · Program_Chairs · 2025-09-17

**Decision:**

Accept (poster)

**Comment:**

This paper proposes a novel Adaptive Classifier-Free Guidance (A-CFG) method, which adaptively constructs unconditional inputs to amplify the divergence between conditional and unconditional logits, thereby enhancing guidance efficacy. The method achieves consistent performance improvements under fixed hyperparameters across multiple datasets, demonstrating effectiveness and stability. However, the authors insufficiently justify the motivation or theoretical foundation for this approach. ​​Notably, the authors have addressed relevant reviewer inquiries with clarity and precision, and these responses have been accepted by the reviewers.​